# Characteristics of microRNAs in Skeletal Muscle of Intrauterine Growth-Restricted Pigs

**DOI:** 10.3390/genes14071372

**Published:** 2023-06-28

**Authors:** Yunhong Jing, Mailin Gan, Zhongwei Xie, Jianfeng Ma, Lei Chen, Shunhua Zhang, Ye Zhao, Lili Niu, Yan Wang, Li Zhu, Linyuan Shen

**Affiliations:** 1Key Laboratory of Livestock and Poultry Multi-Omics, College of Animal and Technology, Sichuan Agricultural University, Chengdu 611130, China; 2021302124@stu.sicau.edu.cn (Y.J.); ganmailin@sicau.edu.cn (M.G.); 2021302137@stu.sicau.edu.cn (Z.X.); 2020202051@stu.sicau.edu.cn (J.M.); chenlei815918@sicau.edu.cn (L.C.); 14081@sicau.edu.cn (S.Z.); zhye@sicau.edu.cn (Y.Z.); niulili@sicau.edu.cn (L.N.); 14916@sicau.edu.cn (Y.W.); zhuli@sicau.edu.cn (L.Z.); 2Farm Animal Genetic Resource Exploration and Innovation Key Laboratory of Sichuan Province, Sichuan Agricultural University, Chengdu 611130, China

**Keywords:** intrauterine growth restriction, pig, skeletal muscle, microRNA, miR-451, C2C12 cells

## Abstract

microRNAs are a class of small RNAs that have been extensively studied, which are involved in many biological processes and disease occurrence. The incidence of intrauterine growth restriction is higher in mammals, especially multiparous mammals. In this study, we found that the weight of the longissimus dorsi of intrauterine growth-restricted pigs was significantly lower than that of normal pigs. Then, intrauterine growth-restricted pig longissimus dorsi were used to characterize miRNA expression profiles by RNA sequencing. A total of 333 miRNAs were identified, of which 26 were differentially expressed. Functional enrichment analysis showed that these differentially expressed miRNAs regulate the expression of their target genes (such as PIK3R1, CCND2, AKT3, and MAP3K7), and these target genes play an important role in the proliferation and differentiation of skeletal muscle through signaling pathways such as the PI3K-Akt, MAPK, and FoxO signaling pathways. Furthermore, miRNA-451 was significantly upregulated in IUGR pig skeletal muscle. Overexpression of miR-451 in C2C12 cells significantly promoted the expression of Mb, Myod, Myog, Myh1, and Myh7, suggesting that miR-451 may be involved in the regulation of the myoblastic differentiation of C2C12 cells. Our results reveal the role of miRNA-451 in regulating myogenic differentiation of skeletal muscle in pigs with intrauterine growth restriction.

## 1. Introduction

Intrauterine growth restriction (IUGR) is defined as the growth rate of a fetus below the normal rate of normal fetal growth potential [1]. The characteristic is that the birth weight of the fetus is less than two standard deviations from the normal birth weight or less than 10% of the normal birth weight [2,3]. IUGR newborn fetus body and organ growth develop slowly, with low resistance and increased perinatal mortality and morbidity [4]. IUGR is the second leading cause of perinatal death, after preterm birth [5]. Fetal growth is a complex process, which is influenced by maternal, fetal, placental, and genetic factors [6,7]. The IUGR incidence rate is 10~15%, which is higher in developing countries [8], making it one of the public health problems of global concern. Research of IUGR shows that the IUGR newborns in human, pig, sheep, and mice are often accompanied by problems with metabolic function [9,10,11], cardiovascular function [12], skeletal muscle development [13,14,15], and the immune system [16,17].

Skeletal muscle accounts for about 40% of fetal body weight [18] and plays a major role in glucose metabolism, insulin resistance, and inflammation [19,20]. Skeletal muscle development is regulated by growth factors, nutrients, and endocrine hormones [21]. Impaired skeletal muscle development is a significant feature of IUGR individuals, and placental dysfunction is the main cause of IUGR [22]. Placental dysfunction leads to an insufficient supply of nutrients and oxygen to the fetus. The body sacrifices part of the energy and oxygen supply of skeletal muscle and the liver to ensure the normal development of the brain [22]. This not only affects the number of muscle fibers but also affects the diameter of muscle fibers, so that the skeletal muscle development of IUGR fetuses is different from that of normal fetuses [23]. The number of skeletal muscle fibers is fixed before birth and no longer changes after occurrence. Therefore, impaired skeletal muscle development will continue into adulthood, even if compensatory growth and development after birth cannot be compensated, and may lead to increased incidence of metabolic diseases such as type 2 diabetes [10,24,25], insulin resistance [25,26], and obesity [27,28].

microRNA (miRNA) is a non-coding RNA with an average length of 22 nucleotides [29,30], and it is also the most reported small RNA. miRNAs are involved in the regulation of target gene expression at the post-transcriptional level by translational inhibition and mRNA destabilization [30,31], and miRNAs have become potential therapeutic and diagnostic targets for various diseases [32]. Studies on miRNA and IUGR also reported that the miR-29 family identified in skeletal muscle of IUGR pigs inhibits cell proliferation and promotes protein degradation by targeting IGF1 and CCND1 [33]. miR-1227-3p may regulate trophoblast cell proliferation and apoptosis by targeting genes involved in the insulin pathway [34]. Another study showed that miR-19a-3p was involved in insulin resistance in IUGR mice [35].

As an important livestock production animal, pigs are also a good disease model animal [36]. Pigs are typical multiparous mammals in livestock, and the incidence of IUGR is 15–25% [37]. Piglets affected by IUGR have slow skeletal muscle development [38], reduced immune function [39], and metabolic disorders [40]. It is very appropriate to study the mechanism and role of IUGR in pigs.

Studies on miRNA characterization of the longissimus dorsi muscle in IUGR pigs after birth have been reported, but miRNAs in the longissimus dorsi muscle of IUGR pigs still have research value for fetal skeletal muscle development. This study characterized miRNA expression in skeletal muscle of normal and IUGR newborn piglets, and the effect of miRNA on fetal skeletal muscle development was analyzed.

## 2. Materials and Methods

### 2.1. Ethics Statement

Animal experiments were performed according to the Laboratory Animal Management Regulations revised by the Ministry of Science and Technology of China in 2004. All experimental animal treatment procedures in this study were approved by the Animal Ethics and Welfare Committee of Sichuan Agricultural University (No. DKY-B20131403).

### 2.2. Animals and Sample Collection

In this study, 9 litters born on the same day were selected for weight analysis, and a total of 18 piglets were selected according to the selection criteria (1 normal pig and 1 IUGR pig per litter according to birth weight). The weight selection criteria for IUGR pigs were two standard deviations below the average birth weight of the herd, and normal pig weight should be controlled within one standard deviation of the average weight. Samples were taken from these pigs at 35 days of age. The left longissimus dorsi (LDM) was weighed, and about 1g of fragments in the middle of the muscle were collected and stored at −80 °C. Three pigs whose weight was the closest to the average body weight of each group were used for small RNA sequencing and reverse transcription quantitative PCR.

### 2.3. Small RNA Sequencing

Total RNA from pig longissimus dorsi was extracted by the Trizol method for small RNA sequencing. We measured the integrity and quantity of RNA samples using Nanodrop 2000 (Thermo, San Jose, CA, USA) and agarose gel electrophoresis, and all RNA samples passed quality control. The results are shown in Appendix A. Some RNA modifications that interfere with small RNA-seq library construction were removed from RNA samples, then ligated to 3′ and 5′ small RNA adapters. cDNA was synthesized and amplified. PCR-amplified fragments were extracted and purified from PAGE gel, and the completed libraries were quantified with an Agilent 2100 bioanalyzer. The sample libraries were denatured and diluted. Sequencing was performed on an Illumina Nextseq 500 system. The sequencing data were processed as we described before [41,42]. Clean data were obtained by removing raw-data adapters and low-quality readings. Clean readings were mapped to pig sequences using miRDeep2. The raw reads are provided in Appendix A. The datasets produced in this study were deposited in the NCBI Sequence Read Archive (SRA), accession number: PRJNA800654. 

### 2.4. Prediction and Functional Annotation of Target Genes

Target genes were predicted using the online platform OmicStudio at https://www.omicstudio.cn/analysis, accessed on 11 February 2023. Genetic ontology (GO) (accessed 14 February 2023) and Kyoto Encyclopedia of Genes and Genomes (KEGG) (accessed 14 February 2023) enrichment analyses were performed on the target genes. Cytoscape software was used to construct a protein–protein interaction and miRNA–target gene–signaling pathway regulatory network. The predicted target genes are shown in Appendix A, and the results of the functional enrichment analysis are available in Appendix A. 

### 2.5. Cell Culture and Transfection 

Cell culture and transfection were conducted according to a previously described method [43]. Briefly, C2C12 myoblasts were cultured at 5% carbon dioxide and 37 °C. The cell proliferation medium was prepared by DMEM (Gibco, Carlsbad, CA, USA) and 10% fetal bovine serum (Gibco). The cell differentiation medium consisted of DMEM and 2% horse serum (Gibco). When C2C12 cell density reached 80%, miR-451 mimic and negative control were transfected into C2C12 myoblasts using Lipo3000 (Ribobio, Guangzhou, China).

### 2.6. Reverse Transcription Quantitative PCR (RT-qPCR) 

The quantification of miRNA and mRNA was performed using the methods in our previous study [33]; the small RNA first strand was synthesized using the Mir-X™miRNA First Strand synthesis kit (Takara, Kusatsu, Japan). RT-qPCR was performed in the Bio-Rad CFX96 Real-Time PCR assay system using TB Green Premix Ex Taq II (Takara). U6 and ACTB were used as internal controls for normalization of the RT-qPCR results of miRNA and mRNA, respectively. The relative expression level was calculated by 2^−ΔΔct^. The primer sequences of miRNAs and mRNAs are shown in Appendix A.

### 2.7. Statistical Analysis

The data were analyzed using Microsoft Excel 2021 and SPSS version 28 (IBM, Armonk, NY, USA). The results were presented as mean ± SD. The data differences between two groups were compared using Student’s *t*-test. The data between the two groups were compared using Student’s *t*-test. *p* < 0.05 was considered statistically significant and *p* < 0.01 was considered as a strong significance.

## 3. Results

### 3.1. Phenotypic Characterization of Pigs with Intrauterine Growth Retardation

The body weight of the IUGR pigs was significantly lower than normal pigs at 14 days post-weaning (35-days) (Figure 1A). The relative weight of longissimus dorsi of the IUGR pigs was also significantly lower than normal pigs at 14 days post-weaning (Figure 1B). Then, we analyzed the correlation between birth weight, body weight at 14 days post-weaning, and longissimus dorsi muscle weight at 14 days post-weaning of IUGR and normal pigs together. We found that the body weight at 14 days post-weaning and longissimus dorsi muscle weight at 14 days post-weaning were significantly positively correlated with birth weight (Figure 1C,D). Furthermore, we can see the differences in body size of IUGR and normal pigs at birth and 14 days post-weaning (Figure 1E). 

### 3.2. The Expression Characteristics of miRNA in IUGR Pig Skeletal Muscle

The expression profiles of miRNAs of porcine skeletal muscle were generated by sequencing. The length of these miRNAs was in range of 19~24 nt, of which those mainly enriched in 21–23 nt accounted for nearly 90% of these miRNAs (Figure 2A). A total of 333 miRNAs were obtained from muscular tissue (Figure 2B). There was an overlap of 89.48% of these miRNAs, with 16 and 19 miRNAs unique to normal and IUGR pigs. Further analyses identified that the miRNA families let-7, miR-30, miR-10, miR-17, and miR-154 were the most numerous members (Figure 2C). The two most abundant miRNAs in pig skeletal muscle were ssc-miR-1 and ssc-miR-206, which represented more than 94% of the total sequences (Figure 2D).

### 3.3. Differentially Expressed miRNAs between IUGR and Normal Pig Skeletal Muscle

The miRNAs with a fold change ≥ 1.5 and *p* < 0.05 were selected as differentially expressed miRNAs (DEMs), which were associated with intrauterine growth retardation. We identified 26 DEMs between normal and IUGR pig skeletal muscles, and among them, 16 were significantly upregulated and 10 were significantly downregulated in IUGR pigs (Figure 2E). The heat map of DEMs demonstrated that the IUGR pig samples clustered together (Figure 2F). Moreover, we analyzed nucleotide bias at seed-sequence positions of significantly upregulated miRNAs (SUMs) and significantly downregulated miRNAs (SDMs) (Figure 2G,H). Twelve randomly selected DEMs (two SUMs and two SDMs) were for RT-qPCR to verify the RNA-seq data. The results of RT-qPCR showed a similar trend to RNA-seq (Figure 2I). Table 1 shows the differentially expressed miRNAs.

### 3.4. Target Gene Prediction and Functional Analysis of Differentially Expressed miRNAs

Several previous studies certified that microRNAs can direct their post-transcriptional inhibitory role by base-pairing to the mRNA 3’-untranslated regions [44]. We predicted the target genes of 16 SUMs,10 SDMs, and the top-2 most expressed miRNAs. A total of 6216, 4242, and 637 target genes had been predicted, respectively. SUMs have 2034 common target genes with SDMs (Figure 3A).

Then, we performed gene ontology (GO) and Kyoto Encyclopedia of Genes and Genomes (KEGG) terms analysis on these predicted target genes. 

The top-two most expressed miRNAs represented more than 94% of the total sequences (Figure 2D) and shared the same seed sequences. We performed the GO enrichment analysis and KEGG terms analysis of them firstly: GO enrichment analysis suggested that the top-two most expressed miRNAs were involved in biological processes including cell proliferation, cell migration, angiogenesis, insulin secretion, and growth (Figure 3C); KEGG enrichment results suggested that the top-two most expressed miRNAs were principally involved in endocytosis, regulation of actin cytoskeleton, and the Ras signaling pathway (Figure 3D).

GO enrichment analysis showed that the SDMs were mainly involved in biological processes, including phosphorylation, cellular response to insulin stimulus, and negative regulation of cell proliferation (Figure 4A), and SUMs were mainly involved in biological processes, including positive regulation of glucose import, phosphorylation, and muscle cell differentiation (Figure 4B). KEGG classification indicated that the SDMs and SUMs were involved in many identical processes (Figure 4C,D). KEGG enrichment results showed that the SUMs and SDMs were primarily involved in insulin resistance, the MAPK signaling pathway, metabolic pathways, and the regulation of the actin cytoskeleton (Figure 4E,F).

### 3.5. Protein–Protein Interaction Network Construction of Target Genes

Protein–protein interactions (PPIs) reflect the previous interactions of a given group of proteins and reveal important principles of protein organization in cells. Therefore, the STRING database was used to construct PPIs that differentially expressed miRNA target genes. It shows the interaction of some target genes (single nodes were removed) (Figure 5). The network consists of 130 nodes and 3554 edges, and GSK3B, AKT3, MAPK14, and EGFR are key proteins in this network.

### 3.6. Regulation Network and Correlation Analysis of Muscle Development-Related Pathways

In order to explore the major regulatory factors related to IUGR skeletal muscle development, we mapped the interaction network by screening the signaling pathways related to skeletal muscle development and associating the genes and miRNAs that might play a role. It is found that miRNA-26a, miRNA-181d-5p, miRNA-296-3p, and miRNA-7-5p have more target genes in the network; PIK3R1, CCND2, AKT3, MAP3K7 CCND3, and CCND1 are the main target genes; and the PI3K-Akt signaling pathway, MAPK signaling pathway, FoxO signaling pathway, and regulation of the actin cytoskeleton are the major enrichment pathways (Figure 6).

### 3.7. miR-451 Is Involved in the Regulation of Skeletal Muscle Differentiation

Studies have shown that miR-451 can be used as a biomarker for cancer [45,46]. In this study, miR-451 was identified as a significantly upregulated miRNA in IUGR pig skeletal muscle. We wondered whether miR-451 affects skeletal muscle growth and development.

Firstly, we quantified the relative expression level of miR-451 in C2C12 cells at different time points of proliferation and differentiation (Figure 7A,B). In the proliferation stage, the expression level of miR-451 was significantly decreased on the second and third days compared with the first day (Figure 7A). In the proliferation stage, the expression of miR-451 increased first, peaked on day 6, and then decreased (Figure 7B). Then, miR-451 was successfully overexpressed in porcine C2C12 cells by transfection of miR-451 mimic (Figure 7C). Overexpression of miR-451 in C2C12 cells significantly promoted the expression of Mb (myoglobin), Myod (myogenic differentiation 1), Myog (myogenin), Myh1 (myosin, heavy polypeptide 1), and Myh1 (myosin, heavy polypeptide 7) (Figure 7D–G).

## 4. Discussion

miRNAs are a class of small non-coding RNAs that play a regulatory role after target gene transcription by inhibiting translation or promoting mRNA deadenylation and attenuation to inhibit protein synthesis [30,47]. miRNAs have been shown to be involved in many biological processes and disease occurrence [48], such as muscle cell proliferation, adipocyte differentiation, senescence, cardiovascular disease, and diabetes.

IUGR is one of the public health problems of global concern [49]. IUGR is typically characterized by lower growth rates and organ development than normal for gestational age, with higher postnatal morbidity and mortality [50] and a greater risk of metabolic, reproductive, immune, and respiratory abnormalities [9,10]. The incidence of IUGR in pigs is 15–25%, causing serious losses to livestock production [51]. The pig is also a model animal for human development and disease research with a high degree of homology with humans. Therefore, pigs are ideal animal models for researching IUGR. In this study, we researched the characterization and function of miRNAs associated with IUGR porcine muscle development, which will contribute to the research of the effects of IUGR on muscle development.

IUGR individuals usually grow slowly and have delayed organ development. In our research, compared with normal pigs, the body weight and growth rate of IUGR pigs at 35 days after birth were significantly lower than that of normal pigs, and the longissimus dorsi index was also significantly lower at 35 days after birth. These results show that our model is correct and can be further studied.

miRNA expression profiles of porcine longissimus dorsi muscle were evaluated by high-throughput sequencing. We identified 333 miRNAs in pig skeletal muscle. miR-1 and miR-206 were the two miRNAs with the highest expression levels, which accounted for more than 94%. These results are consistent with our previous reports [33]. Both miR-1 and miR-206 are members of the miR-1 family and share a common seed sequence. Previous studies have shown that miR-206 expression is significantly upregulated in the process of inducing C2C12 cells to differentiate into muscle tubes [52], and the overexpression of miR-206 in C2C12 cells promotes muscle differentiation [53]. Chen demonstrated that miR-1 promotes myogenesis by targeting histone deacetylase 4 (HDAC4) [54]. Another study demonstrated that ZNF281/ZFP281 can be used as a miR-1 target to counteract muscle differentiation [55]. These results suggest that miRNAs with higher expression abundance may play a role in biological processes.

In this research, a total of 26 differentially expressed miRNAs (DEMs) were identified in IUGR pig skeletal muscle, among which 16 were significantly upregulated and 10 were significantly downregulated. Seven SUM and five SDM miRNAs were randomly verified by RT-qPCR, and the trend of change was consistent with that of RNA-seq, indicating that the sequencing results were reliable. It has been reported that these DEMs are associated with muscle development.

Our published study showed that miR-29a was significantly upregulated in IUGR pig skeletal muscle. In C2C12 cells, overexpression of miR-29a can play a functional role in inhibiting myocyte proliferation by targeting the mRNA-level expression of IGF1 and CCND1 [33]. In adult muscle stem cells, miR-29a mediates fibroblast growth factor-2 (FGF-2) to promote myoblast proliferation, and inhibition of miR-29a expression showed the opposite effect [56]. miR-27a induced insulin resistance in C2C12 skeletal muscle cells by inhibiting PPARγ and its downstream genes involved in the development of obesity [57]. In another study, leucine-induced upregulation of miR-27a inhibited myostatin mRNA expression and proliferation of C2C12 [58]. Samani conducted the in vivo knockout of miR-486 in mice, and mir-486 KO mice showed the overall destruction of muscle fiber structure, decreased cross-sectional area of muscle fiber, and increased fibrosis [59]. Zhang found that circMEF2D regulated the PI3K-AKT signaling pathway by binding to miR-486 and inhibited the proliferation and differentiation of bovine myoblasts [60]. In C2C12 cells, miR-26a can directly target Smad1 and Smad4 to promote differentiation of myogenic cells, whereas the inhibition of miR-26a expression leads to delayed differentiation and skeletal muscle regeneration [61]. miR-101-1 inhibited C2C12 cell differentiation and reduced myotube formation by inhibiting Myod, Myog, and Myhc mRNA and protein expression [62].In addition, miR-486 [59,60], miR-122-5p [63], miR-503 [64,65], miR-7-5p [65], miR-142-5p [66,67], miR-1343 [68], miR-101 [69], miR-21-5p [70], and miR-26a [61] all play important regulatory roles in skeletal muscle development. However, the mechanism of these miRNAs in the skeletal muscle of the IUGR model remains to be further studied.

Most miRNAs inhibit protein synthesis by inhibiting translation [30]. Therefore, we predicted the target genes of significantly downregulated and significantly upregulated miRNAs and performed functional enrichment analysis of the target genes. GO analysis revealed that the biological process of SDMs in IUGR were described as protein phosphorylation, positive regulation of myotube differentiation, negative regulation of cell proliferation, and head morphogenesis; the biological process of SUMs in IUGR were described as positive regulation of glucose import, negative regulation of protein phosphorylation, cell proliferation, and muscle cell differentiation.

KEGG analysis showed that the signaling pathways with DEMs enrichment were highly coincident, mainly enriched in endocytosis, FoxO signaling pathway [71], MAPK signaling pathway [72], regulation of the actin cytoskeleton, metabolic pathways, and insulin resistance [73]. These signaling pathways are closely related to skeletal muscle development. Previous studies have shown that the mTOR signaling pathway [74], insulin resistance [75], and FoxO signaling pathway [76] are related to the occurrence of IUGR. In IUGR mouse experiments, phosphodiesterase 5 inhibitors can promote IUGR fetal growth by enhancing the phosphorylation level of the mTOR signaling pathway and its downstream protein [74]. This suggests that the DEMs we screened are related to IUGR skeletal muscle development.

miR-451 may be involved in regulating IUGR pig skeletal muscle development. Our previous study showed that the muscle fiber diameter of the longissimus dorsi was significantly reduced in intrauterine growth-retarded pigs relative to normally developing pigs [33]. IUGR pigs with lower birth weight had a significantly lower semitendinosus weight and reduced semitendinosus cross-sectional area and myofiber number [23]. Overexpression of miR-22 and miR-499 in C2C12 myoblasts inhibited C2C12 myoblast proliferation and promoted myoblast differentiation into myotubes, while inhibition of miR-22 and miR-499 expression appeared to have the opposite effect [77,78]. Previous research has shown that miR-451 inhibits lipid deposition by targeting the inhibition of acetyl-CoA carboxylase α expression [79]; andmiR-451 has also been widely studied as a cancer biomarker [45]. However, whether miR-451 is involved in the development and regulation of skeletal muscle of IUGR pigs has not been reported.

We quantified miR-451 expression at different developmental stages and time points of C2C12 cell proliferation and differentiation in mice and found that miR-451 continued to express at all stages. We overexpressed miR-451 in C2C12 cells by a miR-451 mimic, and quantified the mRNA of genes related to muscle differentiation. It was found that overexpression of miR-451 could significantly upregulate the mRNA expression levels of Mb, Myod, Myog, Myh1, and Myh7. These results suggest that miR-451 may be involved in the regulation of the myoblastic differentiation of C2C12 cells. Che found that the protein levels of MHC, Myod, and Myog were upregulated by overexpression of miR-1290, which promoted the differentiation of muscle cells [80]. Peng’s study showed that inhibition of miR-133a promoted TGF-β1-induced myoblast differentiation [81]. In addition, numerous studies have shown that miR-451 is involved in the regulation of erythrocyte differentiation [82]. miR-451 was inhibited in a vascular smooth muscle injury model, and Shexiang Baoxin Pill could promote the expression of miR-451, inhibit cell proliferation and migration, and promote apoptosis [83]. In studies of hypertrophic cardiomyopathy (HCM), miR-451 was found to regulate cardiac hypertrophy and cardiac autophagy by targeting tuberous sclerosis complex 1 (TSC1) [84]. It indicated that miR-451 may be involved in the regulatory process of skeletal muscle development in IUGR pigs, but the regulatory mechanism needs to be further investigated.

## 5. Conclusions

In conclusion, the present study provided the expression profiling of miRNAs in the skeletal muscle of IUGR pigs. A total of 26 (16 upregulated, 10 downregulated) differentially expressed miRNAs were identified. Functional enrichment analysis showed that these differentially expressed miRNAs regulate skeletal muscle development in IUGR pigs through signaling pathways such as MAPK, PI3K-AKT, and mTOR. Then, miR-451 was selected for functional verification. We found that overexpression of miR-451 promoted myoblast differentiation of C2C12 cells by significantly upregulating the mRNA expression levels of Mb, Myod, Myog, Myh1, and Myh7. We proved that miR-451 was involved in the regulation of skeletal muscle differentiation development, which is consistent with the results of our functional enrichment analysis. The results of this study provide new evidence for the involvement of miRNAs in IUGR pig skeletal muscle development and provide a basis for future studies on IUGR and skeletal muscle.

## Figures and Tables

**Figure 1 genes-14-01372-f001:**
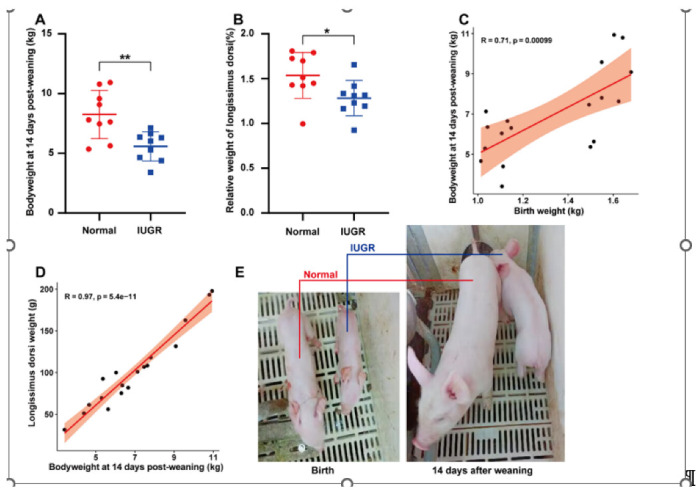
Phenotypic indexes of IUGR and normal pigs. (**A**) The body weight of the IUGR and normal pigs at 14 days post-weaning. (**B**) The relative weight of longissimus dorsi at 14 days post-weaning. (**C**) Correlation with birth weight and body weight at 14 days post-weaning. (**D**) Correlation with weight of longissimus dorsi and body weight at 14 days post-weaning. (**E**) Photograph of IUGR and normal pigs at birth and 14 days post-weaning. (**A**–**D**) Normal group, n = 9; IUGR group, n = 9. ** p* < 0.05, ** *p* < 0.01.

**Figure 2 genes-14-01372-f002:**
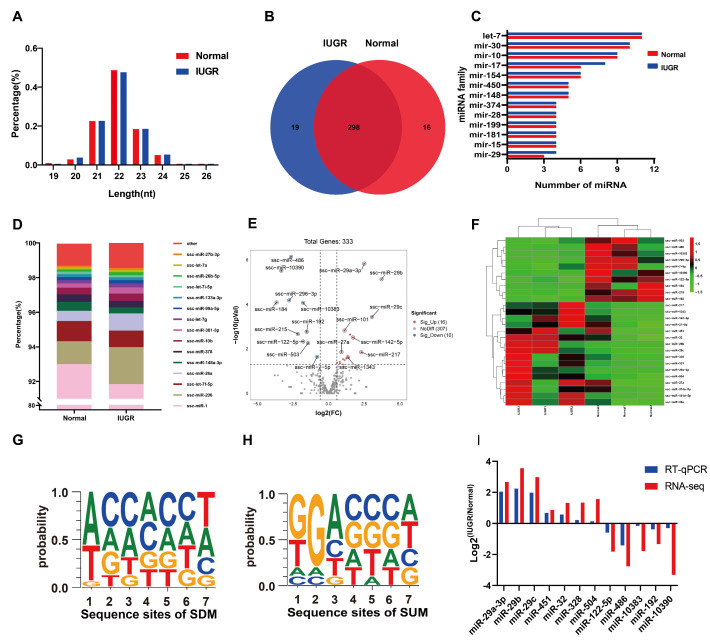
Sequence characteristics of miRNA profiling of pig muscle and identification of differentially expressed miRNAs (DEM). (**A**) Length distribution of miRNA sequences. (**B**) Venn diagram of the number of identified miRNAs. (**C**) The families of the identified miRNAs. (**D**) Composition of the top-15 highly expressed miRNAs. (**E**) The Volcano plot of miRNA expression profiles. Red circles represent upregulated miRNAs in IUGR relative to normal pigs. Blue circles represent downregulated miRNAs in IUGR relative to normal pigs. Gray circles represent no significant differences. (**F**) Heatmap and clustering of DEMs. The basis characteristics of seed sequences for SUM (**G**) and SDM (**H**). (**I**) RT-qPCR validation of partial sequencing results.

**Figure 3 genes-14-01372-f003:**
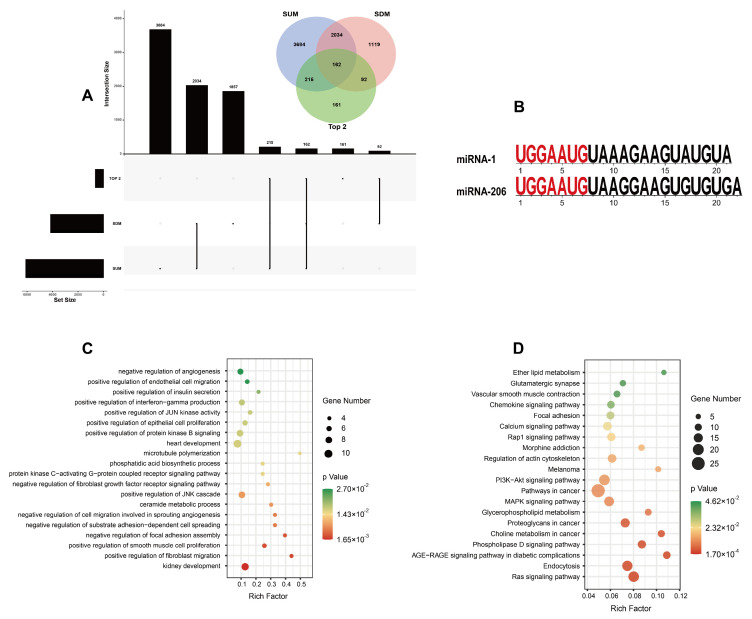
(**A**)The number of predicted target genes for the top-two most expressed miRNAs and differentially expressed miRNAs. (**B**) The sequences of the top-two most expressed miRNAs. Gene ontology (GO) enrichment analysis (**C**) and Kyoto Encyclopedia of Genes and Genomes (KEGG) enrichment analysis (**D**) for the top-two most expressed miRNAs.

**Figure 4 genes-14-01372-f004:**
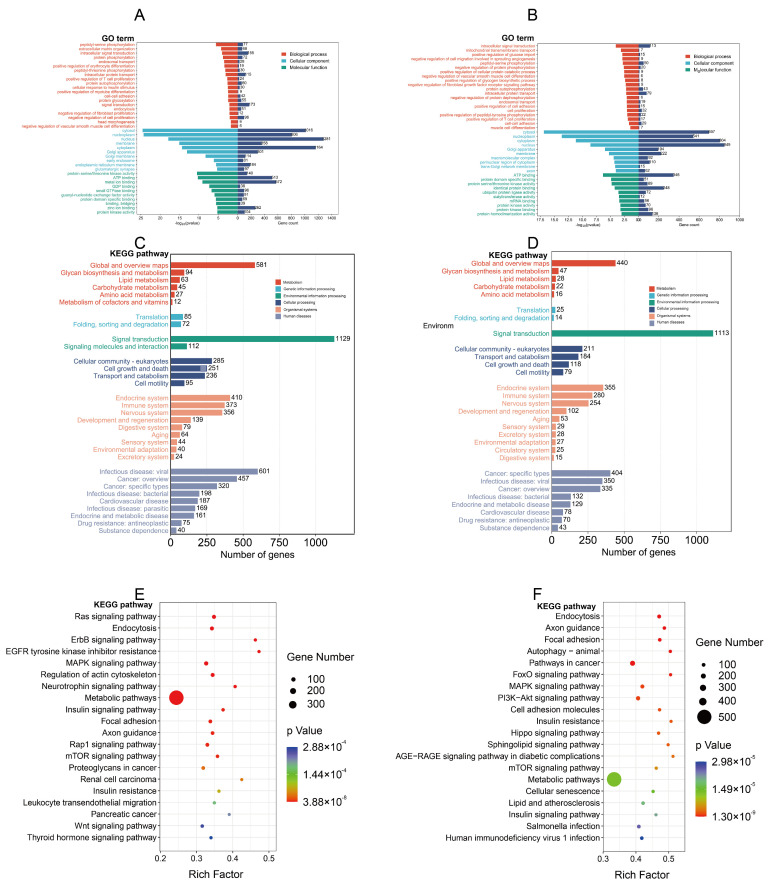
Gene ontology (GO) enrichment analysis and Genomes (KEGG) enrichment analysis. Gene ontology (GO) enrichment analysis for significantly downregulated expressed (**A**) and significantly upregulated expressed miRNAs (**B**). KEGG classifies significantly downregulated expressed (**C**) and significantly upregulated expressed miRNAs (**D**). Genomes (KEGG) enrichment analysis for significantly downregulated expressed (**E**) and significantly upregulated expressed miRNAs (**F**).

**Figure 5 genes-14-01372-f005:**
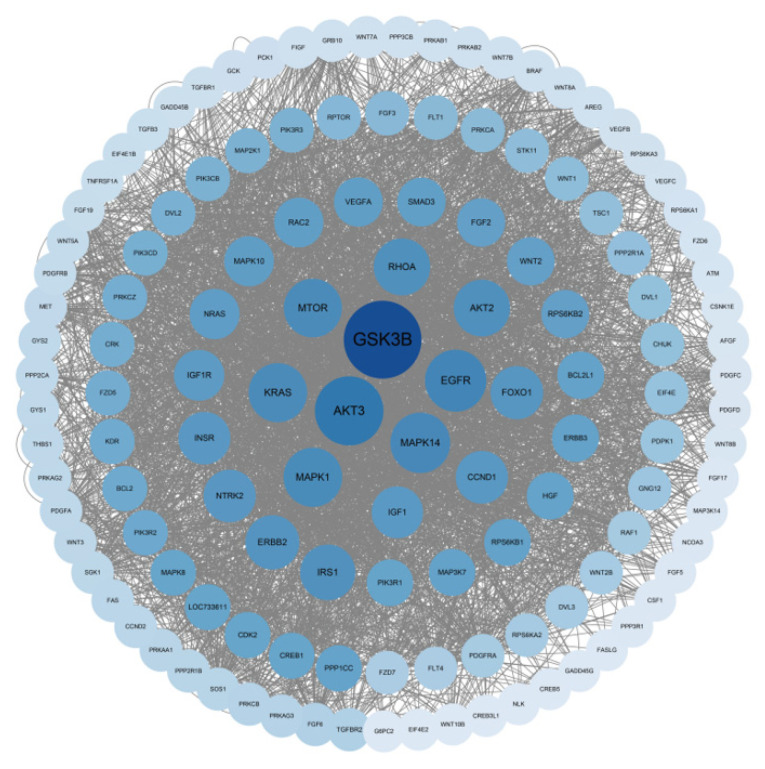
Protein–protein interaction network of target genes of differentially expressed miRNAs. The circle size and color scales represent the node degree.

**Figure 6 genes-14-01372-f006:**
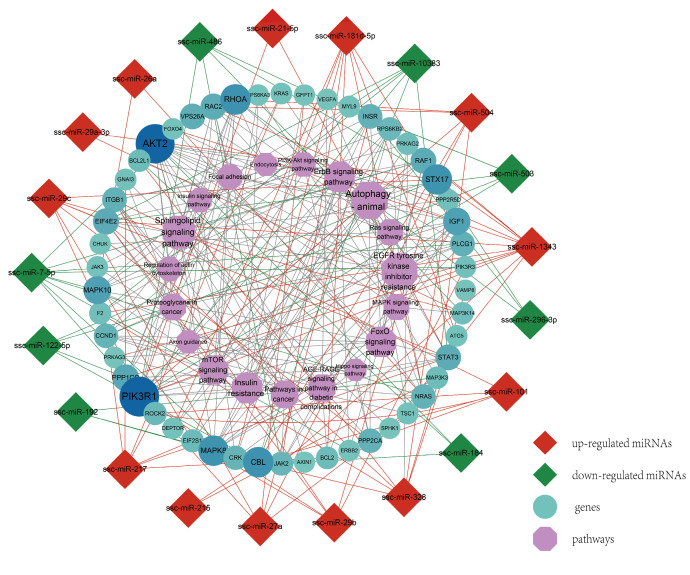
The interaction network of miRNA–target gene–pathway. Hexagonal nodes represent pathways. Circular nodes represent target genes, and the circle size and color scales represent the node degree. Red diamond nodes represent upregulated miRNAs, and green diamond nodes represent downregulated miRNAs.

**Figure 7 genes-14-01372-f007:**
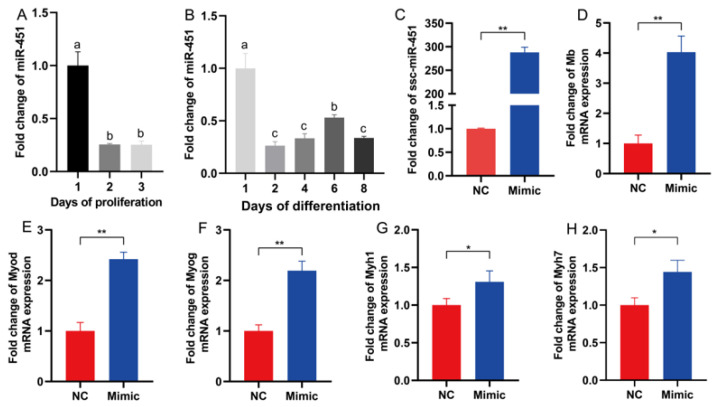
miR-451 is involved in the regulation of gene expression related to skeletal muscle development in C2C12 cells. (**A,B**) The relative expression of miR-451 in the proliferative and differentiation stage of C2C12 cells. (**C**) The relative expression of miR-451 in C2C12 cells after transfection with miR-451 mimic. The relative expression of Mb, Myod, Myog, Myh1, and Myh7 in C2C12 cells after transfection with miR-451 mimic (**D**–**H**). * *p* < 0.05, ** *p* < 0.01.

**Table 1 genes-14-01372-t001:** List of miRNA populations differentially expressed in the spleen between normal and IUGR pigs.

Type	miRNA-ID	Mature_Seq	Normal-CMP	IUGR-CMP	Fold Change	log2FC	*p*-Value
Downregulated	ssc-miR-184	UGGACGGAGAACUGAUAAGGGU	13.777	1.224	0.089	−3.493	0.000082
	ssc-miR-10390	AUACUACUGACAGACCGCAACCU	4.344	0.440	0.101	−3.304	0.000003
	ssc-miR-296-3p	AGGGUUGGGCGGAGGCUUUCC	2.181	0.326	0.150	−2.742	0.000066
	ssc-miR-486	UCCUGUACUGAGCUGCCCCGAG	69.034	10.174	0.147	−2.762	0.000001
	ssc-miR-215	AUGACCUAUGAAUUGACAGAC	4.107	1.039	0.253	−1.983	0.002172
	ssc-miR-10383	UGGUGCCUGACGUCUUGGCAGU	12.415	3.625	0.292	−1.776	0.000086
	ssc-miR-122-5p	UGGAGUGUGACAAUGGUGUUUGU	16.578	4.712	0.284	−1.815	0.004718
	ssc-miR-192	CUGACCUAUGAAUUGACAGCC	14.586	5.810	0.398	−1.328	0.001716
	ssc-miR-503	UAGCAGCGGGAACAGUACUGCAG	10.981	4.610	0.420	−1.252	0.005568
	ssc-miR-7-5p	UGGAAGACUAGUGAUUUUGUUGUU	51.110	31.545	0.617	−0.696	0.023011
Upregulated	ssc-miR-29b	UAGCACCAUUUGAAAUCAGUGUU	0.578	6.817	11.791	3.560	0.000007
	ssc-miR-29c	UAGCACCAUUUGAAAUCGGUUA	0.289	2.283	7.905	2.983	0.000371
	ssc-miR-29a-3p	CUAGCACCAUCUGAAAUCGGUUA	11.353	72.181	6.358	2.669	0.000001
	ssc-miR-217	UACUGCAUCAGGAACUGAUUGGAU	0.143	0.784	5.495	2.458	0.014239
	ssc-miR-142-5p	CAUAAAGUAGAAAGCACUACU	6.530	19.209	2.942	1.557	0.003117
	ssc-miR-504	AGACCCUGGUCUGCACUCUAUCU	3.937	11.668	2.964	1.567	0.002305
	ssc-miR-1343	CUCCUGGGGCCCGCACUCUCGC	0.768	2.116	2.756	1.463	0.026311
	ssc-miR-32	UAUUGCACAUUACUAAGUUGC	0.955	2.391	2.504	1.324	0.023649
	ssc-miR-328	CUGGCCCUCUCUGCCCUUCCGU	2.017	5.117	2.538	1.344	0.026897
	ssc-miR-451	AAACCGUUACCAUUACUGAGUU	55.217	100.369	1.818	0.862	0.032048
	ssc-miR-101	UACAGUACUGUGAUAACUGAA	369.650	896.228	2.425	1.278	0.001511
	ssc-miR-181d-5p	AACAUUCAUUGUUGUCGGUGGGUU	2.722	5.932	2.179	1.124	0.029318
	ssc-miR-27a	UUCACAGUGGCUAAGUUCCGC	49.659	91.698	1.847	0.885	0.014055
	ssc-miR-374a-5p	UUAUAAUACAACCUGAUAAGUG	18.821	33.522	1.781	0.833	0.046061
	ssc-miR-21-5p	UAGCUUAUCAGACUGAUGUUGA	641.470	1073.217	1.673	0.742	0.038817
	ssc-miR-26a	UUCAAGUAAUCCAGGAUAGGCU	5960.309	9830.052	1.649	0.722	0.042270

Note: CPM means counts per million mapped reads, log2FC means log2 (fold change), the positive value of log2FC indicates that the miRNA was upregulated in IUGR porcine skeletal muscle, whereas the miRNA was downregulated in IUGR.

## Data Availability

All data involved in this study are available on https://www.ncbi.nlm.nih.gov/, accession number(s): PRJNA800654.

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
