# Peer review of "Characteristics of microRNAs in Skeletal Muscle of Intrauterine Growth-Restricted Pigs"

_genes, 2023, doi:10.3390/genes14071372_

Round 1

Reviewer 1 Report

Suggestions,

1)    Avoid short forms in abstract. 

2)    Did author check RIN value of RNA? 

3)    Line 87. How sequencing libraries prepared? Information missing.

4)    The paper needs extensive revision for English and grammar. Paper is not well written. Need extensive revision. Consider re-writing.

5)    The paper is not in good shape. Check journal guidelines.

6)    Figure 4. Y axis is not readable.

7)    Consider naming at least 10 significant miRNAs in volcano plot. 

8)    The paper has lots of good results, but paper is not well written. Take help of native English speaker. 

The paper has lots of good results, but paper is not well written. Take help of native English speaker. 

Reviewer 2 Report

The manuscript titled: Characteristics of microRNAs in Skeletal Muscle of Intrauterine Growth-Restricted Pig, is focused on an interesting topic in pig biology that has a wide audience interest and within the scope of this Journal. Indeed, this study applied updated transcriptome profile of miRNAs in growing piglets. In addition, and the authors used the standard format of this Journal. The idea of this investigation sound novel. However, there are some technical clarification of experimental design. In addition, more an additional experiment is needed to improve the significance of data coming out of this investigation.

-Please indicate in details the experimental design of this investigation including the criteria for selection of individuals used for NGS especially IUGR.

-I recommend including an additional experiment to validate the some more of differentially expressed candidate miRNAs this because two main technical reasons (1) the fold change detected and selected for bioinformatics analysis is low 1.5 as well established by scientific community to set 2-fold change as limiting for genome-wide expression data analysis. (2) As clearly shown in the heatmap that there is no stable expression of differentially selected miRNAs however, there is a clear variation within individuals of IUGR and normal groups. 

Round 2

Reviewer 1 Report

The authors have adequately addressed the comments made by me in revised version of the manuscript. Therefore, I have no further comments.

Minor editing of English language required

Reviewer 2 Report

The authors of the manuscript titled: Characteristics of microRNAs in Skeletal Muscle of IntrauterineGrowth-Restricted Pig, have improved the qulity of the manuscript and replied to my queries. Therefore, i recommend acceptance of this manuscript for publication in this outstanding journal.